# Leveraging Parameter Space Symmetries for Reasoning Skill Transfer in LLMs

**Stefan Horoi**[1,2,†,*]**, Sangwoo Cho**[3]**, Supriyo Chakraborty**[3]**,
Shi-Xiong Zhang**[3]**, Sambit Sahu**[3]**, Guy Wolf**[1,2]**, Genta Indra Winata**[3,†]
[1]Université de Montréal    [2]Mila – Quebec AI Institute    [3]Capital One
stefan.horoi@mila.quebec, genta.winata@capitalone.com

**Editors:** Marco Fumero, Clementine Domine, Zorah Lähner, Irene Cannistraci, Bo Zhao, Alex Williams

## Abstract

Task arithmetic is a powerful technique for transferring skills between Large Language Models (LLMs), but it often suffers from negative interference when models have diverged during training. We address this limitation by first aligning the models' parameter spaces, leveraging the inherent permutation, rotation, and scaling symmetries of Transformer architectures. We adapt parameter space alignment for modern Grouped-Query Attention (GQA) and SwiGLU layers, exploring both weight-based and activation-based approaches. Using this alignment-first strategy, we successfully transfer advanced reasoning skills to a non-reasoning model. Experiments on challenging reasoning benchmarks show that our method consistently outperforms standard task arithmetic. This work provides an effective approach for merging and transferring specialized skills across evolving LLM families, reducing redundant fine-tuning and enhancing model adaptability.

## 1 Introduction

The proliferation of open-weight Large Language Models (LLMs) [9, 15, 3, 21] has fostered a decentralized ecosystem of model development. Once a foundational model is released, research teams independently adapt it to diverse applications. However, the rapid release of next-generation base models introduces a developmental dilemma: teams must either continue relying on their older, customized models or incur the substantial cost of re-adapting their work to a newer, more capable version. This challenge underscores the need for methods that can efficiently integrate the specialized skills of an adapted model with the enhanced capabilities of its successor.

A promising strategy for skill transfer is task arithmetic [14], which represents the effect of fine-tuning as a task vector, the difference between the fine-tuned and pre-trained weights. This vector encodes the acquired skill, and by adding multiple task vectors to a base model, their respective skills can be combined [14]. Despite its appeal, task arithmetic is prone to negative interference, where conflicting parameter updates from independently trained models degrade performance [23, 24]. This problem becomes especially severe when models have substantially diverged during adaptation.

This paper addresses two challenges at this intersection: 1) efficiently transferring skills across different versions of a foundational model, and 2) mitigating the negative interference that arises when applying task arithmetic to models with diverging training trajectories. We focus on the crucial skill of LLM reasoning. Cultivating robust reasoning in LLMs is a notoriously difficult and

---

*The work was done while in an internship at Capital One. †Corresponding authors.

Proceedings of the III edition of the Workshop on Unifying Representations in Neural Models (UniReps 2025).

computationally expensive process. Therefore, the ability to transfer these hard-won reasoning skills between model versions is a vital research objective, allowing teams to leverage state-of-the-art advancements without invalidating their prior investments.

**Contributions.** We make several contributions. First, we extend methods that leverage parameter space symmetries, specifically rotation and scale symmetries for attention layers [25] and permutation symmetry for feed forward layers (FFN) [2], to their widely used, modern counterparts, Grouped-Query Attention (GQA) [1] and SwiGLU layers [18]. Second, we use these alignment methods to transfer the reasoning skills of `Nemotron-Nano` [3] from its reference model, `Llama-3.1-8B-Instruct` [9], to the independently instruction-tuned `Tulu3-8B` model. Through extensive evaluations, we show that aligning the parameter spaces before transferring the reasoning skills using task arithmetic significantly improves the final model's performance on hard reasoning benchmarks.

## 2 Preliminaries and Methodology

### 2.1 Task Arithmetic for Skill Transfer

**Task arithmetic** [14] provides an efficient framework for editing model capabilities by performing linear operations directly on their weights ($\theta$). The core idea is to represent a learned skill as a **task vector** ($\tau_{\text{skill}}$), which is the difference between a fine-tuned model's parameters and its original base model's parameters. This vector isolates the changes that encode a specific new skill.

$$\tau_{\text{skill}} = \theta_{\text{fine-tuned}} - \theta_{\text{base}}. \tag{1}$$

This formulation is particularly powerful for **skill transfer**. A skill vector, such as one for reasoning, can be extracted from a reference model pair (e.g., $\tau_{\text{reason}} = \theta_{\text{ref.+reason}} - \theta_{\text{ref.}}$) and then applied to an entirely different target model ($\theta_{\text{target}}$) to imbue it with that same skill. The new model's parameters, $\theta_{\text{target+reason}}$, are synthesized through simple vector addition:

$$\theta_{\text{target+reason}} = \theta_{\text{target}} + \tau_{\text{reason}}. \tag{2}$$

This creates an analogy where the target model is enhanced in the same way as the reference model. While effective, this process can suffer from **negative interference** [23, 24], where conflicting parameter updates between the models degrade performance, especially when their parameter spaces have significantly diverged.

### 2.2 Leveraging Parameter Space Symmetries in Neural Networks

The architecture of modern neural network are characterized by multiple parameter space symmetries, meaning distinct sets of weights can produce an identical output function. For instance, neurons in a feed-forward layer can be reordered, or the internal representations in an attention layer can be rotated, without changing the model's behavior. These misalignment can occur when models are trained from different initializations during diverging training runs

When comparing two independently trained models, these symmetries can cause their parameter spaces to be misaligned, hindering direct arithmetic operations like skill transfer.

#### 2.2.1 Aligning SwiGLU Layers via Permutation Symmetry

The SwiGLU activation function for a given FFN layer in model $i$, $\text{Swish}_\beta(x\mathbf{W}_G^{(i)^\top}) \odot (x\mathbf{W}_U^{(i)})$, uses gate ($\mathbf{W}_G^{(i)}$) and up-projection ($\mathbf{W}_U^{(i)}$) matrices whose intermediate outputs can be permuted without changing the function, provided the down-projection ($\mathbf{W}_D^{(i)}$) is adjusted accordingly. We find the optimal permutation matrix $\mathbf{P}$ that aligns the weights of two models (1 and 2) by solving the linear assignment problem:

$$\underset{\mathbf{P} \in \mathbb{S}_P}{\arg\min} \left( ||\mathbf{W}_G^{(1)} - \mathbf{P}\mathbf{W}_G^{(2)}||^2 + ||\mathbf{W}_U^{(1)} - \mathbf{P}\mathbf{W}_U^{(2)}||^2 + ||\mathbf{W}_D^{(1)} - \mathbf{W}_D^{(2)}\mathbf{P}^\top||^2 \right) \tag{3}$$

$$= \underset{\mathbf{P} \in \mathbb{S}_P}{\arg\max} \langle \mathbf{P}, \mathbf{W}_G^{(1)}\mathbf{W}_G^{(2)^\top} + \mathbf{W}_U^{(1)}\mathbf{W}_U^{(2)^\top} + \mathbf{W}_D^{(1)^\top}\mathbf{W}_D^{(2)} \rangle. \tag{4}$$

This problem is solved efficiently using the Hungarian algorithm. Model 2 weights are then updated:

$$\mathbf{W}_G^{(2)} \to \mathbf{P}\mathbf{W}_G^{(2)}, \quad \mathbf{W}_U^{(2)} \to \mathbf{P}\mathbf{W}_U^{(2)}, \quad \mathbf{W}_D^{(2)} \to \mathbf{W}_D^{(2)}\mathbf{P}^\top. \tag{5}$$

## 2.3 Aligning GQA Layers via Rotation and Scaling Symmetry

The absence of element-wise non-linearities between query, key, and value projections in attention layers allows for continuous rotation and scaling symmetries. We align the GQA layers in three steps.

### 2.3.1 Optimal Rotation for Query and Key Heads

In GQA, all query heads ($Q_k$) in a group ($G_j$) share a single key head ($K_j$). We find a single rotation matrix $\mathbf{R}_{QK_j}$ for this shared space by solving the Orthogonal Procrustes Problem:

$$\underset{\mathbf{R}_{QK_j} \in \mathbb{S}_O}{\arg\min} \left( \sum_{k \in G_j} ||\mathbf{W}_{Q_k}^{(1)} - \mathbf{R}_{QK_j}\mathbf{W}_{Q_k}^{(2)}||^2 + ||\mathbf{W}_{K_j}^{(1)} - \mathbf{R}_{QK_j}\mathbf{W}_{K_j}^{(2)}||^2 \right). \tag{6}$$

The solution is $\mathbf{R}_{QK_j} = U_{QK_j}V_{QK_j}^\top$ from the SVD of $\mathbf{M}_{QK_j} = \sum_{k \in G_j} \mathbf{W}_{Q_k}^{(1)}\mathbf{W}_{Q_k}^{(2)^\top} + \mathbf{W}_{K_j}^{(1)}\mathbf{W}_{K_j}^{(2)^\top}$. The weights are updated as:

$$\forall k \in G_j : \mathbf{W}_{Q_k}^{(2)} \to \mathbf{R}_{QK_j}\mathbf{W}_{Q_k}^{(2)}, \quad \mathbf{W}_{K_j}^{(2)} \to \mathbf{R}_{QK_j}\mathbf{W}_{K_j}^{(2)}. \tag{7}$$

A similar procedure is used to find the optimal rotation $\mathbf{R}_{VO_j}$ for the shared value ($V_j$) and corresponding output ($O_k$) projections.

### 2.3.2 Optimal Scaling for Query and Key Heads

After rotation, we find a scalar $\alpha$ to align the scales of the query and key matrices, denoted $\mathbf{W}_{Q_k}^{(2)'}$ and $\mathbf{W}_{K_j}^{(2)'}$ post-rotation. The functionality is preserved if one is scaled by $\alpha$ and the other by $1/\alpha$. We find the optimal $\alpha$ by minimizing:

$$\underset{\alpha \in \mathbb{R} \backslash 0}{\arg\min} \left( \sum_{k \in G_j} ||\mathbf{W}_{Q_k}^{(1)} - \alpha\mathbf{W}_{Q_k}^{(2)'}||^2 + ||\mathbf{W}_{K_j}^{(1)} - \frac{1}{\alpha}\mathbf{W}_{K_j}^{(2)'}||^2 \right). \tag{8}$$

We expand this equation, differentiate it with respect to $\alpha$ to find the local extrema and multiply by $\alpha^3$ to get rid of denominators. This yields a quartic equation in $\alpha$ that can be solved numerically:

$$\alpha^4 \sum_{k \in G_j} ||\mathbf{W}_{Q_k}^{(2)'}||^2 - \alpha^3 \sum_{k \in G_j} \langle \mathbf{W}_{Q_k}^{(1)}, \mathbf{W}_{Q_k}^{(2)'} \rangle + \alpha \langle \mathbf{W}_{K_j}^{(1)}, \mathbf{W}_{K_j}^{(2)'} \rangle - ||\mathbf{W}_{K_j}^{(2)'}||^2 = 0. \tag{9}$$

The weights are then updated with the optimal $\alpha$:

$$\forall k \in G_j : \mathbf{W}_{Q_k}^{(2)'} \to \alpha\mathbf{W}_{Q_k}^{(2)'}, \quad \mathbf{W}_{K_j}^{(2)'} \to \frac{1}{\alpha}\mathbf{W}_{K_j}^{(2)'}. \tag{10}$$

# 3 Results

## 3.1 Experimental Setup

Our experiment investigates the transfer of specialized reasoning skills across models from parallel development tracks, using 8-billion-parameter models from the Llama 3.1 family. Our setup involves three models originating from a common ancestor, `Llama-3.1-Base`. The first model is the official branch, containing the standard pre-trained `Llama-3.1-Base` and the instruction-tuned `Llama-3.1-Instruct`. The second, a parallel branch, is represented by AI2's Tulu3 series, which was also instruction-tuned from `Llama-3.1-Base` but followed an independent development path. The third group is the skill source: Nvidia's Nemotron model, which was built upon `Llama-3.1-Instruct` to add advanced reasoning capabilities, such as problem deconstruction and self-correction. Our goal is to transfer the specialized reasoning from Nemotron (on the official branch) to Tulu3 (on the parallel branch). To do this, we create a reasoning "skill vector" by subtracting the parameters of `Llama-3.1-Instruct` from those of Nemotron.

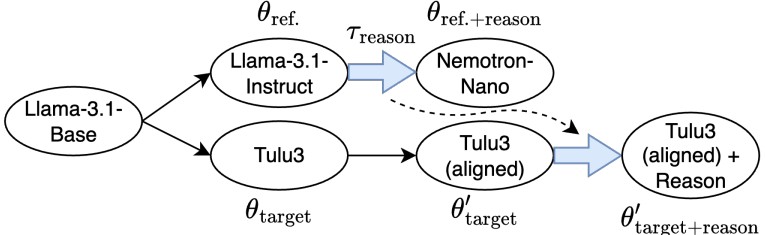

Figure 1: The family tree of models used in our experiments. The diagram illustrates the two divergent fine-tuning branches originating from the common ancestor, `Llama-3.1-Base`. The top branch leads to the reasoning-specialized Nemotron-Nano built on top of `Llama-3.1-Instruct`, while the bottom branch shows the parallel instruction-tuning of the `Tulu3` model. Our work aims to transfer reasoning skills acquired on the top branch to the models on the parallel bottom branch by first aligning the parameter space of `Tulu3` to the one of `Llama-3.1-Instruct`, and then adding the reasoning skill vector.

## 3.2 Parameter Space Alignment Improves Reasoning Skill Transfer

We evaluate the base models, as well as the `Tulu3` model with the reasoning skills transferred (+R) on the following reasoning benchmarks: MATH-500 [12], Minerva-MATH [16], Olympiad-Bench [10], AMC 2023, AIME 2024 and TheoremQA [4]. We compare adding the reasoning skill vector $\tau_{\text{reason}}$ to the `Tulu3` model with and without parameter space alignment to the reference `Llama-3.1-Instruct` model. We consider two ways of computing the optimal permutations, rotations and scaling factors, one based on the model activations (`act.align`) and one based on the model weights (`w.align`). We report average results over 8 different random seed evaluations and the standard errors in table 1. We add more details about evaluations in the Appendix.

Table 1: Accuracy and standard error (%) on difficult reasoning benchmarks of the default models and our Tulu3 + Reasoning (with and without neuron matching) models.

| Model | MATH-500 | Minerva-MATH | OlympiadBench | AMC23 | AIME24 | TheoremQA | Avg. |
|---|---|---|---|---|---|---|---|
| Llama-3.1-Instruct | 45.9±0.39 | 21.9±0.58 | 14.1±0.00 | 28.8±0.72 | 3.3±0.00 | 23.9±0.59 | 30.8 |
| Tulu3 | 39.6±0.45 | 15.4±0.34 | 16.1±0.30 | 24.7±1.92 | 4.6±0.88 | 23.6±0.20 | 29.3 |
| Nemotron-Nano-v1 | 91.4±0.34 | 53.7±0.44 | 60.7±0.19 | 92.5±0.67 | 58.4±1.99 | 54.2±0.29 | 69.7 |
| Tulu3+R | 83.4±0.51 | 42.3±1.00 | **58.8**±0.28 | 87.2±1.45 | 55.9±0.56 | 49.0±0.45 | 63.3 |
| Tulu3(act.align)+R | **85.8**±0.41 | **43.4**±0.79 | 57.3±0.30 | 85.6±0.91 | 56.7±1.79 | **50.2**±0.30 | 63.8 |
| Tulu3(w.align)+R | 83.8±0.21 | 42.5±0.78 | 57.5±0.37 | **90.0**±0.67 | **61.2**±1.40 | 49.2±0.30 | **64.4** |

Firt we observe that the two IF models perform poorly on these complex reasoning tasks, especially when compared to `Nemotron-Nano`. Secondly, with a simple task arithmetic approach we can successfully transfer some of the reasoning skills from `Nemotron-Nano` to the `Tulu3` model, with `Tulu3+R.` performing significantly better than the base `Tulu3`. Lastly, aligning the parameters of `Tulu3` to those of `Llama-3.1-Instruct` improves the reasoning skill transfer even further, with `Tulu3(act.align)+R` and `Tulu3(w.aligned)+R` outperforming `Tulu3+R` on most tasks. Surprisingly, our `Tulu3(w.aligned)+R)` model even outperforms `Nemotron-Nano` on the difficult AIME24 benchmark. We include results on other benchmarks in the Appendix.

## 4 Conclusion

In this work, we tackle the important challenge of reasoning skill transfer in LLMs, demonstrating that parameter space alignment significantly improves the process. Our main contribution is extending recent alignment methods to support modern architectures with Grouped-Query Attention (GQA) and SwiGLU layers. By aligning models prior to skill transfer via task arithmetic, we show performance gains on difficult reasoning benchmarks. For future work, we plan to extend this analysis to other model families and skills beyond reasoning.

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

# A  Experimental details

All evaluations of the models were ran on one p4d.24xlarge node with 8xA100 GPUs with 40GB of VRAM each. We used versions of EleutherAI's LM Evaluation Harness [7] for the evaluation on non-reasoning tasks and the Math Evaluation Harness [8] for reasoning tasks. We evaluate the models on the reasoning tasks with temperature of 1.

**Data used for activation-based alignment**   To align the `Tulu3` model parameter space to that of `Llama-3.1-Instruct` using activations, we feed the model the prompts from the following datasets: MMLU (validation set) [11], DROP (validation set) [6], MATH (dev. set) [13], GSM8K (training set) [5], Tulu-3 SFT Personas Code (train set) and Tulu-3 WildChat IF On Policy (train set) [15]. We only extract the activations of the input prompts, stopping the activation extraction before generation starts.

# B  Ablation study

We also perform an ablation to see which symmetries make the biggest impact when accounted for during the matching process, the results are shown in table 2. First we note that in our experiments, the optimal permutations found through the weight alignment process are always the identity (i.e. no neuron permutations). We hypothesize this is because the models haven't diverged enough during their separate training procedures to warrant the need for a hard re-ordering of any neurons. We see that the biggest gain in accuracy comes from accounting for rotations. Scaling the $Q$ and $K$ layers also brings some minor improvements beyond the non aligned reasoning transfer model.

Table 2: Accuracy and standard error (%) on difficult reasoning benchmarks for different symmetries taken into consideration

| Model | MATH-500 | Minerva-MATH | OlympiadBench | AMC23 | AIME24 | TheoremQA | Avg. |
|---|---|---|---|---|---|---|---|
| Tulu3+R | 83.4±0.51 | 42.3±1.00 | 58.8±0.28 | 87.2±1.45 | 55.9±0.56 | 49.0±0.45 | 63.3 |
| Tulu3(act.align rot.)+R | 84.1±0.45 | 42.7±1.22 | 58.1±0.31 | 91.9±1.22 | 58.7±2.43 | 48.7±0.24 | 64.5 |
| Tulu3(act.align scale)+R | 84.0±0.64 | 40.9±0.56 | 58.7±0.39 | 87.5±1.83 | 57.9±1.40 | 49.3±0.45 | 63.6 |
| Tulu3(w.align)+R | 83.8±0.21 | 42.5±0.78 | 57.5±0.37 | 90.0±0.67 | 61.2±1.40 | 49.2±0.30 | 64.4 |

# C  Evaluations on other benchmarks

We also evaluate our models on other benchmarks: IFEval [26], MMLU Pro [22], GPQA [17], MUSR [19] and BBH [20]. The results are shown in table 3. We use greedy sampling for these benchmarks, therefore we report only a single run of evaluations. We note a drop in accuracy on these tasks when transferring reasoning skills to the `Tulu3` model. This is somewhat expected since task arithmetic approaches are known to suffer from negative interference leading to worst performance on some tasks than the performance of the original models [14, 23, 24]. We note that aligning the `Tulu3` parameter space to that of `Llama-3.1-Instruct` doesn't cause any further drops in performance on these benchmarks, instead we observe minimal gains. Interestingly, it seems that activation alignment (`Tulu3(act.align)+R`) specifically behaves in a different fashion than both no alignment (`Tulu3+R`) and weight alignment (`Tulu3(w.align)+R`). `Tulu3(act.align)+R` preserves the strong instruction following skills from the base models while performing worse on some tasks such as MMLU Pro and BBH.

Table 3: Accuracy (%) on non-reasoning benchmarks.

| Model | IFEval | MMLU Pro | GPQA | MUSR | BBH | Avg. |
|---|---|---|---|---|---|---|
| Llama-3.1-Instruct | 79.3 | 36.4 | 32.7 | 37.8 | 48.5 | 47.0 |
| Tulu3 | 83.5 | 28.3 | 29.5 | 43.0 | 48.0 | 46.4 |
| Nemotron-Nano-v1 | 79.0 | 31.9 | 31.1 | 33.9 | 46.0 | 44.4 |
| Tulu3+R | 59.7 | 29.2 | 27.4 | 37.3 | 41.3 | 39.0 |
| Tulu3(act.align)+R | 80.1 | 12.5 | 28.3 | 40.7 | 35.0 | 39.3 |
| Tulu3(w.align)+R | 60.1 | 29.2 | 27.9 | 37.3 | 41.3 | 39.2 |

