# OpenReview forum: "Leveraging Parameter Space Symmetries for Reasoning Skill Transfer in LLMs"
_NeurIPS.cc/2025/Workshop/UniReps — UniReps2025_

### Official Review · Reviewer_uSju · 2025-09-12

**Confidence:** 4

**Review:**

The paper tackles skill transfer between independently fine-tuned LLMs, focusing on reasoning. The core idea is to align parameter spaces before task arithmetic so that adding a “reasoning skill vector” (difference between a reasoning model and its base) interferes less when applied to a target model. Concretely, the authors extend symmetry-based alignment to modern Transformer components: (i) permutation alignment for SwiGLU FFNs, and (ii) rotation and Q/K scaling alignment for GQA attention layers. They then transfer reasoning skills from Nemotron-Nano (built on Llama-3.1-Instruct) to Tulu3-8B, finding consistent gains over plain task arithmetic. However, the performance on non-reasoning tasks degrades.


## Strengths

- Clear, well-motivated problem: Effective skill transfer across diverged model branches is important for practical deployment and research iteration. The paper defines the negative-interference issue with task arithmetic and proposes a principled fix.


- Methodological novelty for modern stacks: Adapting symmetry alignment to GQA and SwiGLU makes the approach applicable to current LLM families. The formulations are well-specified and easy to reimplement.


- Empirical evidence on hard reasoning:  MATH-500, Minerva-MATH, OlympiadBench, AMC23, AIME24, and TheoremQA, alignment improves over naive transfer (Tulu3+R), and weight-aligned Tulu3 sometimes outperforms Nemotron-Nano (e.g., AIME24).




## Weaknesses

- Narrow transfer setting: Experiments focus on one source–target pair within the Llama-3.1 8B family (Nemotron-Nano, Tulu3). It’s unclear how well the approach generalizes across architectures (e.g., Mistral, Gemma), parameter scales, and other variations.

- Limited baseline comparisons: It would be natural to benchmark against stronger model-merging baselines (e.g., TIES-Merging, other interference-aware or layer-wise merging schemes) and adapter composition/merging alternatives.


- Drop on non-reasoning benchmarks: While expected, the drops on non-reasoning benchmarks remain a practical limitation. The paper does not explore mix-in strategies (e.g., partial-layer merging, weight ensembling gates, low-rank orthogonalization, or per-task scaling factors) to mitigate this.

## Conclusion

The paper offers a practical improvement to task arithmetic by aligning parameter spaces, with solid gains on reasoning benchmarks. The main limitations are the considerable drop on non-reasoning benchmarks, scope (single source-target pair), and  baseline coverage (missing stronger merging baselines)

**Score:**

3

**Topic Fit:**

3

---

### Official Review · Reviewer_qYts · 2025-09-15

**Confidence:** 4

**Review:**

Review:

The paper investigates the concept of extracting task vectors in LLMs and investigates their efficacy for transfer of specialized skills across LLMs.

Pros:

1. The work is a relatively original contribution in task vector interpretability.
2. The motivation and methodology are clearly explained along with relevant experimental results.

Cons:

1. It is unclear whether the “reasoning” task vector is relevant for reasoning- the benchmarks used for evaluation include mainly math benchmarks.  In Appendix C, some other evaluations like BBH show drop and many tasks in BBH are exclusively geared to reasoning. If the task vector for reasoning leads to unstable behaviour across different reasoning tasks, then it is unclear what kind of utility this vector has.
2. The significance of this work in light of the scope of the workshop is unclear. We do not observe any attempts to investigate convergence of representations across models/systems.

While the work starts on an interesting premise, it needs more rigorous grounding in related literature and more experiments on what task vectors mean or how they converge across models to be of interest to the community.

**Score:**

2

**Topic Fit:**

1

---

### Official Review · Reviewer_V2bX · 2025-09-16
**Official Comment**

**Confidence:** 3

**Review:**

The paper tackles the challenge of transferring specialized reasoning skills across independently fine-tuned Large Language Models (LLMs), where naive task arithmetic often introduces negative interference due to diverged training trajectories. The authors extend parameter space alignment methods by accounting for permutation symmetries in SwiGLU layers and rotation and scaling symmetries in Grouped-Query Attention (GQA). They evaluate the transfer of reasoning skills from Nemotron-Nano, a reasoning-specialized model, to Tulu3, an independently instruction-tuned model. Experiments on challenging reasoning benchmarks (MATH-500, Minerva-MATH, OlympiadBench, AMC 2023, AIME 2024, TheoremQA) show that aligned models consistently outperform naive task arithmetic and, in some cases, even surpass the reasoning-specialized source model. Ablation studies indicate that rotation symmetries contribute the most to effective alignment. However, the method introduces trade-offs: while reasoning transfer improves, performance on non-reasoning tasks declines.
## Strength:
1. The paper advances parameter space alignment techniques by adapting them to modern LLM components (SwiGLU, GQA), making the approach directly applicable to current architectures.
2. Rigorous evaluations demonstrate that alignment substantially improves reasoning skill transfer compared to naive task arithmetic, even enabling surpassing the performance of the reasoning-specialized model on some benchmarks.
3. The ablation study disentangles the contributions of permutation, rotation, and scaling symmetries, showing that rotations are the most impactful, adding interpretability to the method.
## Weakness:
1.  Experiments focus only on derivatives of the Llama-3.1 family (Nemotron, Tulu3), leaving questions about the method’s generalizability across more diverse model families.
2. Weight alignment permutations defaulting to the identity (lines 380-382) suggest that the studied models may not have diverged substantially, leaving uncertainty about performance when models are more strongly misaligned.
3. While reasoning is central, exploring other impactful skills (for example, coding or multimodal reasoning) would strengthen the generality and appeal of the approach.
## Comments:

1. Terms like “model version” (line 32) are underspecified, making it ambiguous whether the reference is to type, size, or successor variants.
2. The statement “These misalignment can occur … training runs” (lines 61–62) lacks direct evidence or citations.

**Score:**

4

**Topic Fit:**

3

---

### Official Review · Reviewer_M5ue · 2025-09-17
**Review of the paper "Leveraging Parameter Space Symmetries for Reasoning Skill Transfer in LLMs"**

**Confidence:** 3

**Review:**

The paper proposes a simple yet elegant solution to infuse LLMs with new capabilities without retraining by utilizing the concept of task vectors. However, as the authors observe, a simple addition of task vectors might result in harmful interference, especially when the original model and the transformed model have diverged. To address this limitation, they transform the model's parametric space using standard algebraic manipulation of scaling and rotation on the model weights and activations. This transformation is shown to be beneficial in incorporating reasoning capabilities into a non-reasoning model and outperforms a naive addition of the reasoning task vectors.
The proposed approach is generalizable and reduces reliance on redundant fine-tuning to enhance the models' adaptability.

The only weakness in this work is that the approach shows inconsistent performance when the model activations are compared to the model weights; a detailed analysis of these two types of transformation and their comparative performance would further strengthen the paper. Additionally, while the primary focus of the task is to infuse reasoning capabilities into non-reasoning/general-purpose models, it appears that this method does not work for non-reasoning benchmarks, often resulting in a decrease in model performance, as shown in Table 3. More analysis on why this is the case should also be highlighted. Finally, the performance gains overall from all of this alignment seem incremental, and thus statistical tests are necessary to confifm whether the alignment improves performance significantly or not.

Finally, a few comments and typographic errors in the paper:

1. Lines 66-67 - The symbols for the gate and up-projection matrix are interchanged in the Swish formula.
2. The term aligned model in Figure 1 and in this paper feels a bit misleading since alignment has different connotations these days, i.e., preference tuning. It might be better to use a different verb, such as 'transformed' or 'updated' instead.

**Score:**

3

**Topic Fit:**

2